# Performance of 3D-Printed Continuous-Carbon-Fiber-Reinforced Plastics with Pressure

**DOI:** 10.3390/ma13020471

**Published:** 2020-01-19

**Authors:** Jun Zhang, Zude Zhou, Fan Zhang, Yuegang Tan, Yiwen Tu, Baojun Yang

**Affiliations:** 1School of Mechanical and Electronic Engineering, Wuhan University of Technology, WuHan 430070, China;junzhang_918@126.com (J.Z.); zudezhou@whut.edu.cn (Z.Z.); ygtan@whut.edu.cn (Y.T.); ywtu_whut@163.com (Y.T.); 2Chongqing Huashu Robotics Co., Ltd., FoShan 528234, China; baojunyang_huashu@126.com

**Keywords:** continuous carbon fiber, pressure, composites, 3D printing, FDM

## Abstract

Fused Deposition Modeling (FDM) has been investigated as a low-cost manufacturing method for fiber-reinforced composites. The traditional and mature technology for manufacturing continuous-carbon-fiber-reinforced plastics is Automated Fiber Placement (AFP), which uses a consolidation roller and an autoclave process to improve the quality of parts. Compared to AFP, FDM is simple in design and operation but lacks the ability to pressurize and heat the model. In this work, a novel method for printing continuous carbon-fiber-reinforced plastics with a pressure roller was investigated. First, the path processing of the pressure roller was researched, which will reduce the number of rotations of the pressure roller and increase the service life of the equipment and the efficiency of printing. Thereafter, three specimens were printed under different pressures and the tensile and bending strength of specimens were tested. The tensile strength and bending strength of specimens were enhanced to 644.8 MPa and 401.24 MPa by increasing the pressure, compared to the tensile strength and bending strength of specimens without pressure of 109.9 MPa and 163.13 MPa. However, excessive pressure will destroy the path of the continuous carbon fiber (CCF) and the surface quality of the model, and may even lead to printing failure.

## 1. Introduction

Carbon fiber reinforced plastics (CFRPs) have been widely used in the fields of automobile manufacturing and aerospace due to their excellent mechanical properties such as light weight, high strength, etc. [1,2]. Automated Tape Laying (ATL) and Automated Fiber Placement (AFP) are two commonly used technologies for CFRP manufacturing at present [3,4]. However, the technologies cannot be popularized in civilian applications such as the sports industry (carbon fiber bicycles, carbon fiber badminton rackets etc.) because of the expensive equipment and the complex manufacturing process. The challenge of manufacturing CFRPs is combining the reinforcement fibers into the plastics with the orientation control of fibers and low cost [5]. 

Fused Deposition Modeling (FDM) has developed rapidly due to its low production cost and the high degree of automation in recent years [6,7]. This technology can manufacture complex parts with various thermoplastic filaments such as Polylactic Acid (PLA), Acrylonitrile Butadiene Styrene (ABS), and Nylon, etc. However, one limitation of FDM is the poor mechanical properties of parts [8,9]. The strength of parts printed by FDM is so low that the parts can’t be used directly, the parts are used as a mould to validate the rationality of the design model usually. Wang et al. added thermally expandable microspheres to diminish voids between deposition lines on the basis of FDM [10]. It proves that adding the thermally expandable microsphere to the thermoplastic filaments can improve the mechanical properties of parts fabricated by FDM. Sithiprumnea et al. studied the influence of exfoliated graphene nanoplatelets (xGnPs) on the ABS. The elastic modulus and dynamic storage modulus of 3D-printed parts along three different build orientations were increased by the presence of xGnPs in the ABS matrix [11]. Kalappa et al. fabricated the conductive polymer nanocomposites based on FDM and analyzed their mechanical, electrical and electromagnetic-induction-shielding properties of the nanocomposite [12]. Much work has been done to study the properties of composites based on FDM, the obvious strength improvement of reinforced material is combining the thermoplastics with Carbon Fiber (CF). Table 1 presents an overview of different studies on printing CFRPs, showing the reinforcement of CF on thermoplastics.

The reinforcement of CF on thermoplastics can be summarized into two ways: combining with short carbon fiber (SCF) and combining with continuous carbon fiber (CCF). The effect of SCF on thermoplastics is based on the orientation, dispersion, and void formation of fibers. The bridging and pulling-out of fibers in thermoplastics can increase the strength of PLA, ABS, and PVA composites. The interfacial interaction and glass-transition temperature of short-carbon-fiber-reinforced plastics (SCFRPs) are related to the CF content [13,14,15]. Karsli et al. investigated the effects of fiber length content on the mechanical, thermal, and morphological properties of CF reinforced PA6 composites [16]. The results revealed that the tensile strength of PA6 composites increased from 44 MPa to 86 MPa as the CF content increased to 20%. Li et al. investigated the effects of SCF length on polytetrafluoroethylene (PTFE), and found that the geopolymer matrix strength increased from 33.8 to 85 MPa as the length of SCF increased to 80 um [17]. Ashori et al. studied the improvement in mechanical of polypropylene (PP) reinforced by SCF and xGnPs [18]. The xGnPs can further increase the strength of SCF–PP composites. However, it was found that the properties of parts printed by SCFRPs were slightly better than pure thermoplastic, while additional porosity and poor bonding can be detected because of the presence of SCF [19]. With a continuing increase in fiber length from 80 um, the deformation of SCF decreased, and the gaps between the SCF and PTFE increased gradually [17].

The ideal solution to achieve better properties of parts by FDM is printing with continuous-carbon-fiber-reinforced plastics (CCFRPs) [20]. The orientation of CCFs in composites is easily controlled and CCFs have a smaller specific surface area than SCFs, which increases the interfacial interaction of the CCFs with the thermoplastics. According to current research, two approaches—real-time 3D printing of CCFs with the thermoplastic and printing CCFRPs based on prepared CCFRPs—are applied to manufacturing CCFRPs using FDM. As shown in Figure 1a, the CCFs were impregnated with thermoplastic in the nozzle and the CCFRPs can be printed in a single step [21]. The CCFs and thermoplastic are compounded inside the nozzle, the CCFs are brought out while the thermoplastic is extruded depending on the viscosity of the thermoplastic. Based on this method, Nanya et al. tested the performance of CCFRPs. The results indicated that CCF has a significant effect on the printing model performance and the preferable bonding interfaces were achieved of CCFRPs [22]. Tian et al. studied the influencing of process parameters on the interfaces and performance of printed composites. The strength of printed composites increases with increasing the content of CCF [23,24]. However, CCFs and thermoplastic resin are mixed at the melting zone of the extrusion head, which will cause the non-uniform distribution of CCFs in the resin and the insufficiency of compounding due to the not enough melting zone. As a result, the reinforcement of CCFs to resin is weakened to some extent.

The other strategy is to manufacture the CCFRPs and then print parts with the CCFRPs on a FDM printer, which is a multi-step process, as shown in Figure 1b. The CCFs and thermoplastic are compounded into CCFRPs on a specific material-preparation machine that can control the diameter and fiber content of CCFRPs. Then, the prepared CCFRPs are used to print the parts on the 3D printer, and the process parameters of the 3D printer need to match the continuity characteristics of the CCFRPs. From the perspective of composites, Liviu et al. investigated the influence of Poission’s ratio of different material combinations, and proposed a new formula to estimate the singularity order for the elasto-plastic stress singularity [28]. Wu et al. presented the fabrication method as well as the microstructural and dielectric characterization of bespoke composite filaments for fused deposition modeling (FDM) 3D printing of microwave devices [29]. The method may be used as a guide for the successful fabrication of many types of composite filament with varying functions for a broad range of applications. Goh et al. provide a database of the mechanical properties of additively manufactured polymeric materials fabricated using material extrusion to identify the research gaps [30]. Jahangir et al. studied the reinforcement of material extrusion 3D printed polycarbonate using continuous carbon fiber [31]. Hu et al. analyzed the effect on the performance of the model caused by printing time, printing speed, and layer thickness based on this method [25]. The results indicated that the flexural strength and flexural modulus of printed composites significantly improved with the proposed method with specified printing parameters. Melenka et al. evaluated and predicted the elasticity of the fiber-reinforced 3D-printed structures by MarkOne, which is a 3D printer made by Markforged Company for printing CCFRPs [26]. Dickson et al. assessed the performance of continuous carbon-, Kevlar-, and glass-fiber-reinforced composites by MarkOne [27]. However, 3D printers are simple in design and use but lack the ability to apply additional pressure and heat to the part compared to the ATP machines [32]. Zhang et al. proposed a post-heading and pressurizing on parts printed with Nylon and CCFRPs [33]. This process increases the interfaces in the parts printed by CCFRPs but complicate the manufacture of CCFRP parts. Without pressure, the CCFRPs between adjacent layers are adhered by the thermoplastic during printing, CCFRPs molecular chains of two adjacent layers do not diffuse, and the degree of diffusion will affect the binding strength of the parts. For the manufacture of CCFRP parts, pressure is essential. Lee et al. presented a model in the manufacturing process of semicrystal line thermoplastic matrix composites and studied the effect of pressure on the bonding of two adjacent layers of composites [34]. Wool et al. presented a theory of crack healing in polymers and how the surface rearrangement stage affects the diffusion initiation function and topological features of the interface [35]. Kim et al. presented a microscopic theory of the diffusion and randomization stages based on the reptation model of chain dynamics by de Gennes and proved that the interlayer bonding strength is proportional to the molecular chain diffusion [36]. Song et al. presented the interlaminar intimate contact model and the molecular chain diffusion mode and the experimental results found the degree of the interlaminar intimate could reach 1 when the pressure of the roller reaches 1500 N [37]. The effects of pressure on CCFRPs in the above studies are all based on using AFP to manufacture CCFRPs, but there is less research on the preparation of CCFRPs parts on 3D printing.

To solve the above problem, we propose a novel 3D printing structure for CCFRPs that can heat and pressurize the CCFRPs with a pressure roller during printing. Using this process, the CCFRPs are pressurized while printing to increase the bond between layers. The path processing of the pressure roller is designed and verified to ensure the feasibility and rationality of the algorithm. Then, the testing of printing CCFRPs is performed and the mechanical properties of parts under pressure are analyzed. In this paper, a pressure function is added to 3D printing for CCFRPs, and a matching pressure-roller-path processing algorithm is studied. The results show that pressure brings an obvious improvement to the strength of parts printed using CCFRPs. This article will contribute a new reference and data to the application of 3D printing for CCFRPs.

## 2. 3D Printing of CCFRPs

### 2.1. The Principle of Printing CCFRPs

Figure 2 is a schematic of 3D printer for CCFRPs. The printing process of CCFRPs is completed under the machine coordinate system of seven movements (three movements of the nozzle: X, Y, Z; four movements of CCFRPs: F (feeding), C (cutting), R (rotating), L (lifting). The nozzle can move by a processed path on the X, Y, and Z axis. The printing machine can finish the printing process of the part with the movements of the nozzle and the feeding movement (F), while there is no jumping point in the path of the part. However, jumping points inevitably exist in the path of a complex model. This will limit the application of printing CCFRPs if the handling of jumping points is ignored [29]. Therefore, the cutting process (C) has been added to the printing. Furthermore, other types of 3D printing for CCFRPs lack the ability to apply additional pressure and heat to the part compared to the ATP for CCFRPs used in the current work. The pressure roller we use here is adopted to increase adhesion between layers and the bonding of fibers in one layer. The implementation of pressure can be decomposed into two movements: a rotating movement (R) and a lifting movement (L). The pressure roller is rotated at a certain angle according to the path of the nozzle to ensure that the CCFRPs during printing are pressed. The pressure roller is lifted to a suitable height before rotating to prevent the CCFRPs from being crushed. Then the pressure roller is dropped to continue to pressurize the part when the rotation is completed. In addition, a heating device is installed in the pressure roller to keep the CCFRPs in a molten state for better properties of the parts. 

### 2.2. Path Processing of Pressure Roller

There are two kinds of paths (the path of nozzle and the path of pressure roller) in the molding process of printing CCFRPs. The path of the nozzle is planned by the sliced G-code based on the model. The path of the pressure roller is the same as the nozzle when the path of nozzle is a straight line. The path of the pressure roller is different from the nozzle when the path of the nozzle is a curve. The 3D printing machine implements movements according to the G-code and the curves consist of many points with short distances in the G-code. The pressure roller will lift and rotate frequently if the path is a curve, which will reduce the service life of the device and efficiency of printing. To solve the above problem, the curved path of pressure roller has been converted to a straight-line path. The flow chart of path processing of the pressure roller is shown in Figure 3. Firstly, the paths of pressure roller are divided into line and curve. Then the curve is transformed to a line according to the algorithm is shown in Figure 4. Finally, rotating and pressing codes based on the lined path are added to complete the path processing of the pressure roller.

The flow chart for converting the curve of pressure roller to a line is shown in Figure 4a. The result of this conversion is shown in Figure 4b. The solid line is the curved path of the pressure roller, the dotted line is the straight-line path of the pressure roller after conversion, and the red area is the rolling area of the pressure roller after conversion. The curves are inside the red area, indicating that the rolling is sufficient. The steps for converting curves of pressure roller into lines are as follows:

First, the starting point $\left(X_{n},Y_{n}\right)$ and the ending point $\left(X_{m},Y_{m}\right)$ of curve are found.

Second, the maximum distance (L) of the line $l((X_{n},Y_{n}),(X_{i},Y_{i}))$ with the curve $c((X_{n},Y_{n}),(X_{i},Y_{i}))$, i=n+1 are calculated.

Third, the size of L is compared to (D/2−1); i gradually increases. D is the diameter of the pressure roller. 

Fourth, the $c((X_{n},Y_{n}),(X_{i},Y_{i}))$ is converted to $l((X_{n},Y_{n}),(X_{i},Y_{i}))$ while L>=(D/2−1), and n=i. 

Fifth, the steps above are followed to convert the curve to a line starting from $\left(X_{i},Y_{i}\right)$. 

A model that includes lines and curves was adopted to test the path processing of pressure roller. In Figure 5, (a) shows the model, (b) shows the path of the nozzle, (c) shows the path of the pressure roller, (d) shows the path of the nozzle and the pressure roller when printing the bounding box of the model. The pressure roller will rotate six times ($\theta_{1},\theta_{2},\theta_{3},\theta_{4},\theta_{5},\theta_{6}$) in a build circle of the path and the curve has been converted to three lines. The area of the red line is the pressurized area of the pressure roller, which indicates that the CCFRPs can be pressurized completely.

## 3. Equipment and Experiment 

### 3.1. The Equipment for Printing CCFRP

As shown in Figure 6a, the 3D printer for printing CCFRPs consists of its structure, hardware, and a computer. The computer sends instructions, the hardware accepts instructions, and the structure executes instructions. Figure 6b shows the core components that perform the function of feeding, cutting, rotating, and pressurizing. Figure 6c, shows he prepared CCFRPs of which the diameter is 0.58 mm. Figure 6e shows the details of the nozzle and the pressure roller. The pressure roller can lift and drop accurately under the control of a ball screw, and the pressure can be adjusted by setting the distance of the pressure roller lower than the nozzle. Furthermore, two heating rods are used to heat the nozzle and the pressure roller in the device. The heat in the nozzle can soften the polymeric matrix in the CCFRPs so that the CCFRPs can be easily deposited, and the heat in the pressure roller can keep the matrix in a molten state when pressurizing the part, which makes the adhesion between the layers better.

### 3.2. Printing Process Parameters

Printing the CCFRPs is different from printing the PLA. The feeding speed must be consistent with the nozzle moving speed to avoid the accumulation and breaking of CCFRPs and the speed must be slow to ensure the rolling effect of pressure roller when printing the CCFRPs, the feeding speed and the moving speed is 20 mm/min. The material diameter is 0.58 mm; layer thickness is set to 0.2 mm to increase adhesion between layers. In this paper, 1k carbon fiber is used to prepare the composite material, and the fiber content is 10.3%. Setting the printing parameters is crucial for printing CCFRPs and CURA is used to slice the part in this article. The multi-process parameters of printing are specified as the Table 2.

### 3.3. Path Experiment 

Figure 7 shows the four states of the pressure roller when printing a build circle of paths with CCFRPs. The path of the pressure roller is consistent with the path shown in Figure 5c and the angles of rotation of the pressure roller are the same as the angles shown in Figure 5d. The pressure roller has applied pressure on all CCFRPs whether the path of pressure roller is a curve or a line. The experiment explains the correctness and rationality of the path processing of the pressure roller.

### 3.4. Strength Test

The tensile test model is designed and printed according to the standard GB/T1040.1-2006. The size diagram of the sample for the tensile test is shown in Figure 8a. The length of the sample is 150 mm, the width is 15 mm, the thickness is 2 mm and the tensile speed is 5 mm/min. The CCFRPs are closely aligned in the lengthwise direction. The printed sample of the tensile test is shown in Figure 8b. Figure 8c shows the test machine. The bending test was carried out in the form of three-point bending, according to the standard GB/T449:2005. The size of the sample for the three-point bending test is shown in the Figure 8d. The length is 80 mm, the width is 15 mm, the thickness is 3 mm, the bending speed is 5 mm/min, and the three-point bending length is 50 mm. The CCF is closely aligned along the length of the sample and is perpendicular to the direction of the bending. The printed sample is shown in Figure 8e. Figure 8f,g shows the results of the tensile and bending strength test.

## 4. Results and Discussion

### 4.1. Pressure Path Analysis

The schematic diagram of the pressurized printing of CCFRPs is shown in Figure 9c. The distance between the nozzle and the platform is 0.2 mm so that the CCFRPs stay continuous and adhere to the platform during printing. The CCFRPs are laid on the platform by their own weight and the adhesion between the thermoplastic and the platform. The part printed without pressure is shown in Figure 9a. The CCFRPs in one layer are bonded together by PLA. There are gaps between the CCFRPs without applying pressure on part, resulting in reducing adhesion of CCF. The pressure roller is lower than the nozzle and can press the parts. Pressure allows the CCF to collapse and expand, extruding the excess PLA from the CCFRPs and applying the excess PLA on the expanded CCF. Figure 9b shows the part printed in accordance with the path shown in the Figure 5. This process can increase the cross-linking of CCF and the strength of the model, and the validity of the algorithm of pressure roller is verified. Pressurization is a compaction process that can reinforce the cross-linking of inter-laminar CCFRPs and the adhesion of the adjacent layer of CCFRPs, so the strength of model can be increased. 

### 4.2. Tensile Strength and Bending Strength Analysis

The results of the tensile test and the bending test of the models printed by PLA, CCFRPs, CCFRPs1, CCFRPs3, and CCFRPs5 are shown in Figure 10. The model printed with CCFRPs1 means that the pressure roller is 0.1 mm lower than the nozzle when printing the model. The model printed with CCFRPs3 means that the pressure roller is 0.3 mm lower than the nozzle when printing the model. The model printed with CCFRPs5 means that the pressure roller is 0.5 mm lower than the nozzle when printing the model. The tensile strength of the model printed with PLA is only 42.76 MPa. The tensile strength of the model printed with CCFRPs is 109.9 MPa, which shows that the strength of PLA is improved by CCF. However, the reinforced effect is not obvious and cannot exert the excellent reinforced performance of CCF. The three models printed under different pressures are tested to analyze the effect of pressure on strength. The strength of the model printed with CCFRPs1 is 291.56 MPa, and the strength is increased by 682% compared to PLA. It means the pressure is crucial in the printing of CCFRPs and can enhance the cross-linking and adhesion of CCFs inside the model. Pressure can enhance printing greatly and ensure the excellent reinforced performance of the CCFs. The strength of the models printed by CCFRPs3 and CCFRP5s are 491.76 MPa and 644.8 MPa, representing increases of 1150% and 1508% compared to PLA, respectively. It means that the strength of the model printed with CCFRPs can be enhanced by increased pressure, and the effect is obvious. 

The bending strength of the model printed with PLA is 68.64 MPa. The bending strength of the model printed with CCFRP is 163.13 MPa—an increase of 238% compared to PLA. The bending strength of the models printed with CCFRP1, CCFRP3, and CCFRP5 are 256.76 MPa, 357.43 MPa, and 401.24 MPa, representing increases of 374%, 521%, and 585% compared to PLA, respectively, as shown in Figure 10. The above data show that the bending strength of models after pressurization is significantly improved and the greater the pressure, the better the effect of CCF reinforcement. The results show that the error of each experiment is within 10%, which proves that the experiment is repeatable. 

However, excessive pressure will dislocate the CCFs. As shown in Figure 11, the model printed with CCFRPs5 is deformed due to excessive pressure. The CCF is pulled up due to excessive pressure, which destroys the surface quality of model. Furthermore, the model cannot be formed when we increase the pressure of the pressure roller based on printing the model with CCFRPs5. Although the strength is increased, the surface quality and dimensional accuracy of model is reduced. 

## 5. Conclusions

FDM has been investigated as a low-cost manufacturing method for fiber-reinforced composites and there are many studies on this subject currently. For the manufacture of CCFRP parts, pressure is essential. Without pressure, the CCFRPs between adjacent layers are adhered by the thermoplastic during the printing, CCFRP molecular chains of two adjacent layers do not diffuse, and the degree of diffusion will affect the binding strength of the parts. The traditional and mature technology for manufacturing CCFRPs is AFP, which uses a consolidation roller and an autoclave process to improve the quality of part. Compared to AFP, FDM is simple in design and operation but lacks the ability to apply pressure and heat to the model. This work proposed a molding process for printing CCFRPs to combine the advantages of FDM and AFP. A novel 3D-printing structure for CCFRPs is put forward in this article, which can heat and pressurize the CCFRPs with a pressure roller during printing. Using this process, the CCFRPs are pressurized while printing to increase the bond between layers. Then the strength of printed model with this molding process is tested. The details are as follows:Seven movements (X, Y, Z, F, C, R, L) in the molding process of printing CCFRP are coordinated to complete the printing. The pressure roller is rotated and lifted to apply pressure and heat to the part.The path of pressure roller is sometimes different to that of the nozzle. Any curved path of pressure roller needs to be converted to a straight path to prevent jittering and to improve efficiency. Then, the rotation angle of the pressure roller is calculated based on the straight path. The accuracy of path processing for pressure roller has been proved by the experiment. The phenomenon shows that pressure can enhance the CCF bonding.The tensile and bending strength of models printed by CCFRPs with different levels of pressure have been tested to analyze the effect of pressure. The results show that pressure has a significant effect on the reinforcement of CCF on PLA. The tensile strength and bending strength of the model are enhanced to 644.8 MPa and 401.24 with increased pressure. The results show that the error of each experiment is within 10%, which proves that the experiment is repeatable. However, excessive pressure will destroy the path of the CCFs and the surface quality of model, and may even lead to printing failure.

It can be seen that pressure is necessary for printing CCFRP with FDM. This work will provide the technical foundations for printing CCFRPs with FDM.

## Figures and Tables

**Figure 1 materials-13-00471-f001:**
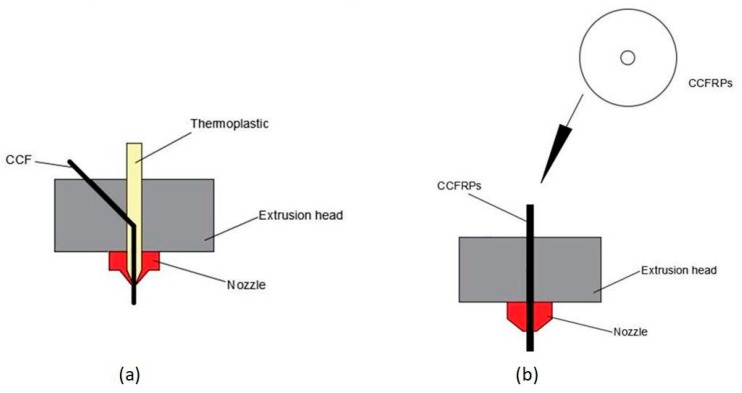
(**a**) The real-time 3D printing of CCF with thermoplastic and (**b**) printing CCFRPs based on prepared CCFRPs.

**Figure 2 materials-13-00471-f002:**
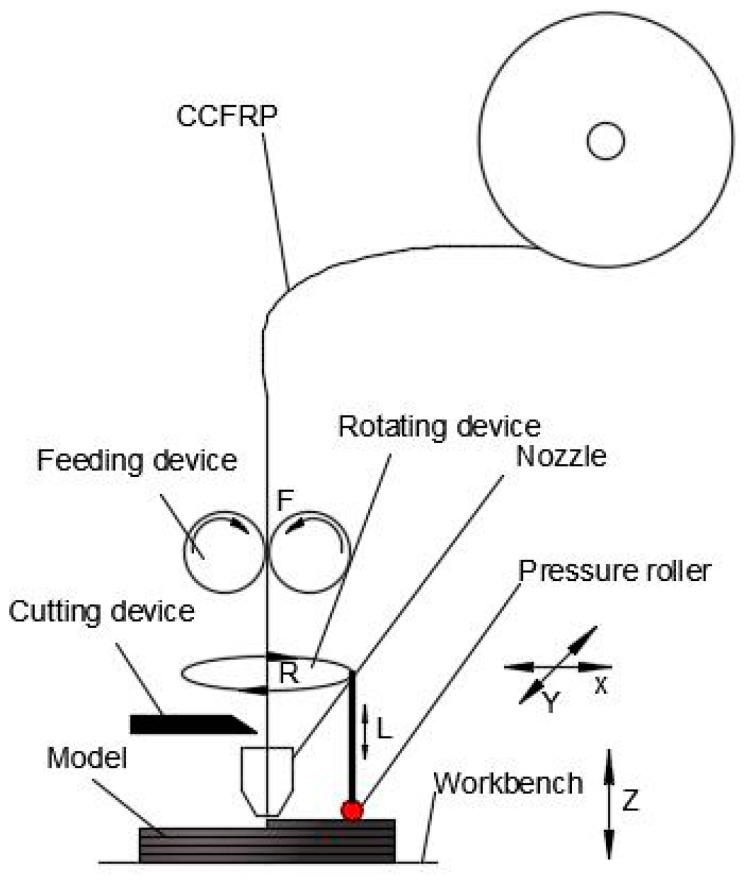
Schematic of 3D-printing machine for CCFRP.

**Figure 3 materials-13-00471-f003:**
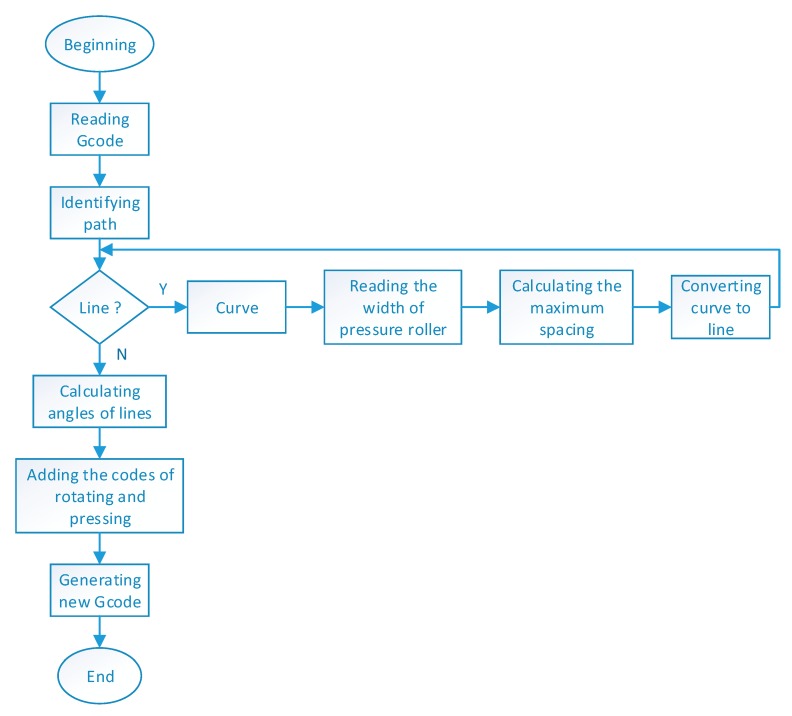
The flow chart for path processing of the pressure roller.

**Figure 4 materials-13-00471-f004:**
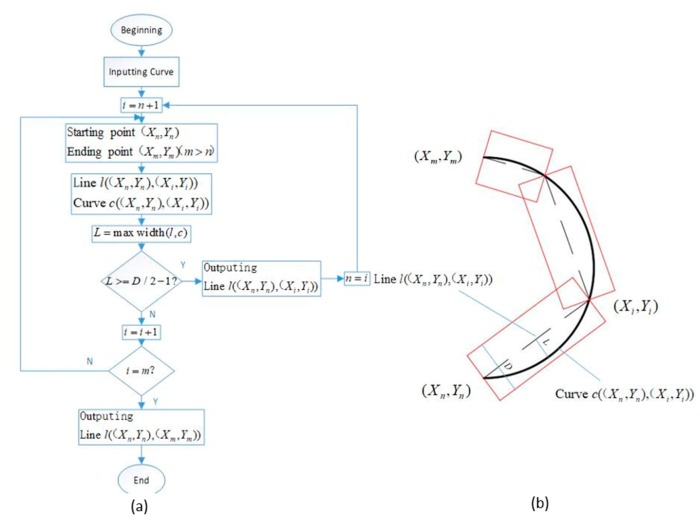
(**a**) The flow chart for converting curve to line and (**b**) the result of converting curve to line.

**Figure 5 materials-13-00471-f005:**
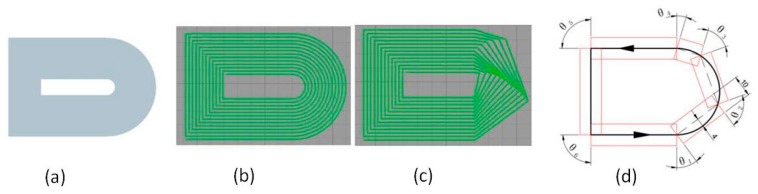
(**a**) The printed model, (**b**) the path of the nozzle, (**c**) the path of the pressure roller and (**d**) the path of the nozzle and pressure roller in the outermost circle of the model.

**Figure 6 materials-13-00471-f006:**
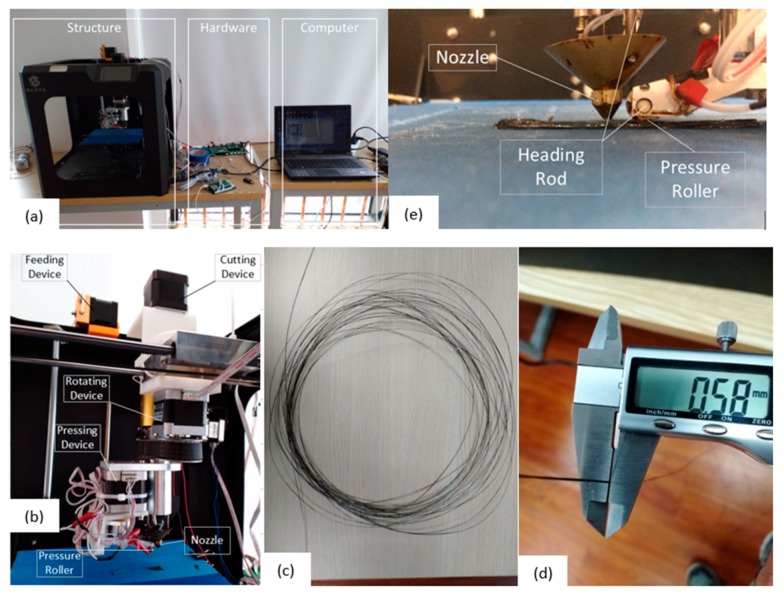
(**a**) The equipment for printing CCFRPs, (**b**) the nozzle device, (**c**) the prepared CCFRPs, (**d**) the diameter of CCFRPs and (**e**) the pressure roller.

**Figure 7 materials-13-00471-f007:**
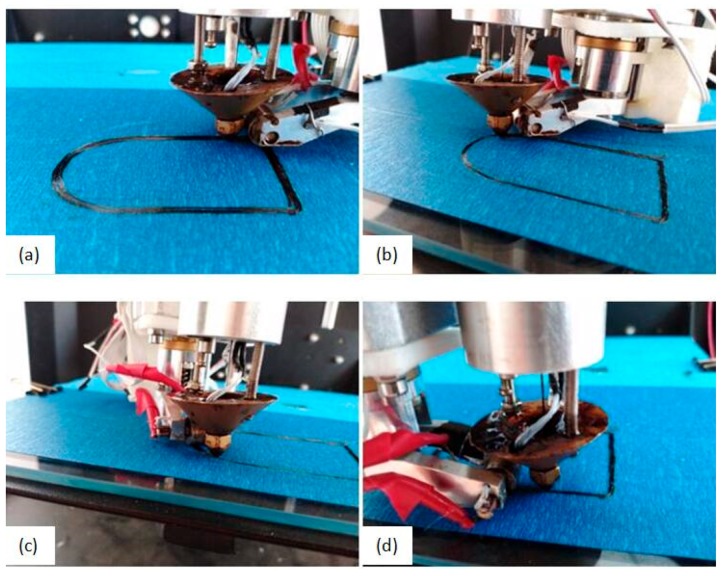
The four states of the pressure roller when printing a circle of paths. (**a**) state before rotation; (**b**) state after $\theta_{1}$ rotation; (**c**) state after $\theta_{3}$ rotation; (**d**) state after $\theta_{4}$ rotation.

**Figure 8 materials-13-00471-f008:**
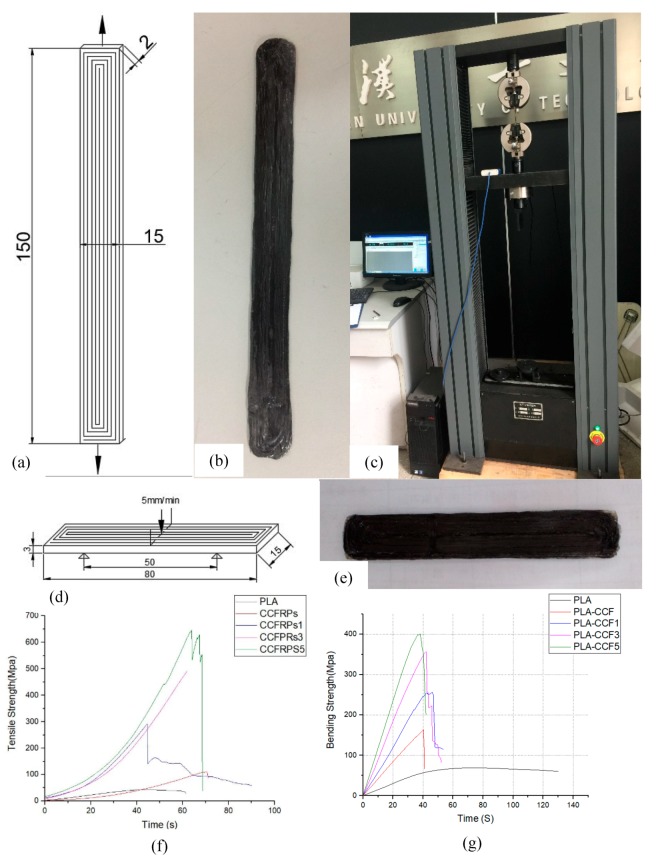
(**a**) The model of the tensile test (unit: mm), (**b**) the printed sample of the tensile test, (**c**) the test machine, (**d**) the model of the bending test, (**e**) the printed sample of the bending test, (**f**) the result of the tensile test, and (**g**) the result of the bending test.

**Figure 9 materials-13-00471-f009:**
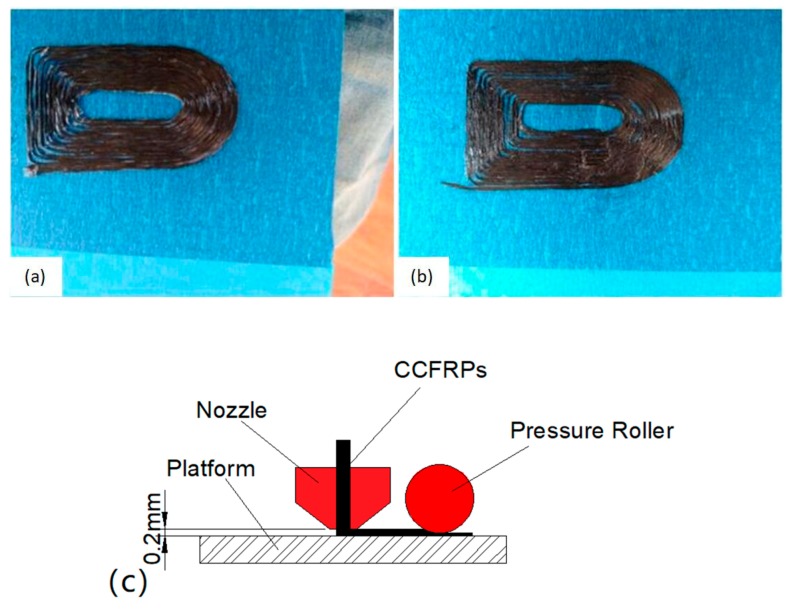
The printing model without pressure (**a**) and with pressure (**b**), and (**c**) the schematic diagram of the pressurized printing CCFRPs.

**Figure 10 materials-13-00471-f010:**
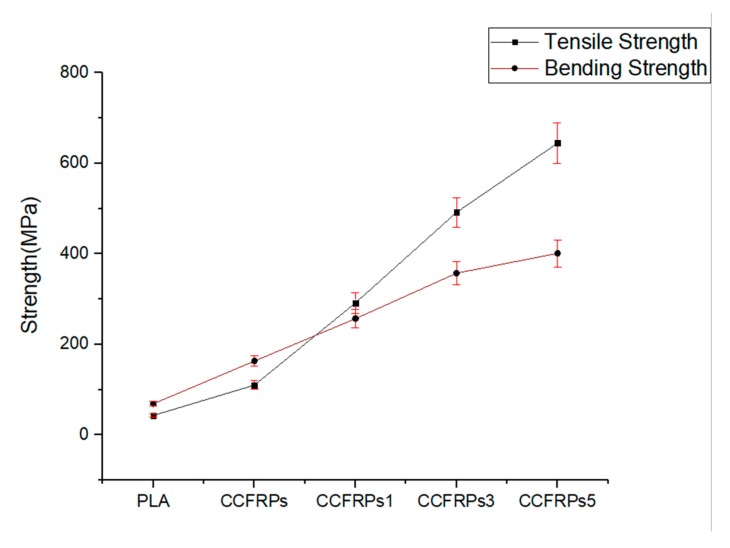
The tensile strength and bending strength of parts under different pressures.

**Figure 11 materials-13-00471-f011:**
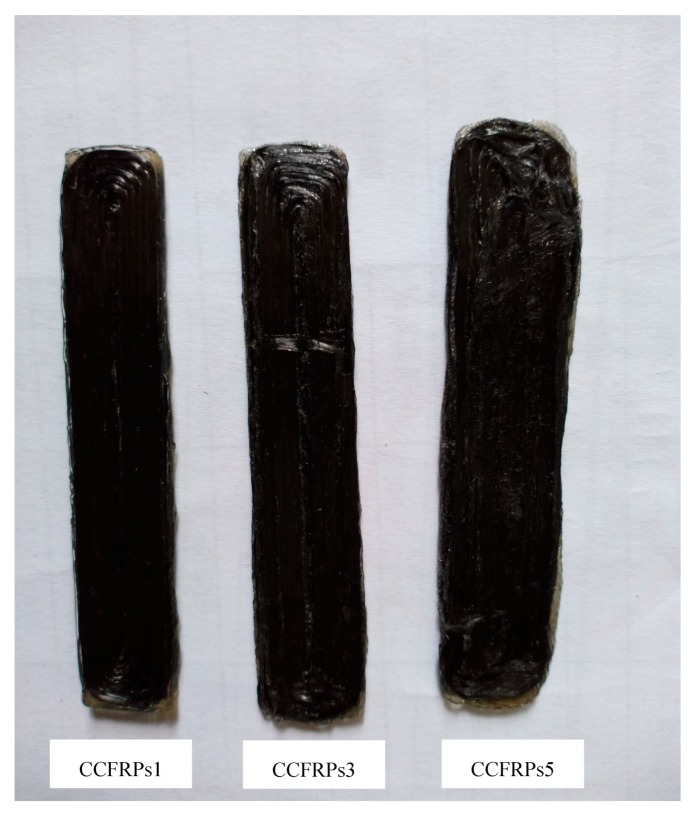
The models printed by CCFRPs1, CCFRPs3 and CCFRPs5.

**Table 1 materials-13-00471-t001:** The overview of studies on printing of reinforced filaments.xGnPs: graphene nanoplatelets, SCF: short carbon fiber, CCF: continuous carbon fiber, CKF: continuous Kevlar fibre, CGF: continuous glass fibre, ABS: acrylonitrile butadiene styrene, PLA: polylactic acid, PU: poly urethane, PVA: polyvinyl alcohol, PA6: polyamide 6, PTFE: polytetrafluoroethylene, PP: polypropylene.

Overview of Studies on Printing of Reinforced Filaments			
Study	Reinforcement	Matrix	Tensile Strength Reinforcement
Sithiprumnea Dul. [11]	xGnPs	ABS	Strength from 33.4 MPa to 38.2 MPa
Tekinalp HL et al. [13]	SCF	ABS	Strength from 35 MPa to 65 MPa
Xiu H et al. [14]	SCF	PLA and PU	Strength from 68.3 MPa to 100 MPa
Istvánné Ráthy et al. [15]	SCF	PVA	Strength from 39 MPa to 51 MPa
Karsli NG et al. [16]	SCF	PA6	Strength from 38 MPa to 86 MPa
Li ZH et al. [17]	SCF	PTFE	Strength from 33.8 MPa to 85 MPa
Ashori A et al. [18]	SCF	PP	Strength from 38.2 MPa to 43.4 MPa
Quan Z et al. [19]	SCF	ABS	Strength from 3.14 MPa to 4.9 MPa
Bettini P et al. [20]	CCF	PLA	Strength from 34 MPa to 203 MPa
Matsuzaki R et al. [21]	CCF	PLA	Strength from 40 MPa to 185 MPa
Li N et al. [22]	CCF	PLA	Strength from 28 MPa to 91 MPa
Tian X et al. [23]	CCF	PLA	Strength from 110 MPa to 335 MPa
Yang C et al. [24]	CCF	ABS	Strength from 30 MPa to 200 MPa
Hu Q et al. [25]	CCF	PLA	Strength from 74.2 MPa to 541.7 MPa
Melenka GW et al. [26]	Continuous Kevlar Fiber	Nylon	Strength from 5 Pa to 92 Pa
Dickson AN et al. [27]	CGF, CCF, CKF	Nylon	Strength from 61 Pa to 444 Pa

**Table 2 materials-13-00471-t002:** The parameters of printing CCFRPs.

Parameter	Amount
Nozzle diameter	0.8 mm
Material diameter	0.58 mm
Feeding speed	20 mm/min
Moving speed	20 mm/min
Rotating speed	2 rad/s
Layer thickness	0.2 mm
Infill percentage	100%
Fiber content	10.3%
Extrusion temperature	200 °C
Pressure roller temperature	180 °C

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
