# Peer review of "Performance of 3D-Printed Continuous-Carbon-Fiber-Reinforced Plastics with Pressure"

_materials, 2020, doi:10.3390/ma13020471_

Round 1
Reviewer 1 Report
The subject matter of the paper is within the scope of the journal and has a good technical quality.
The article draws attention to the influence of pressure on the 3D printing for continuous Carbon fibre reinforced plastics.
AUTHORS:
- The title is generic. Please modify the title.
- Please write lines 143-147.
- The Figure 3 and 4 are not clear. Author, please give a better quality to Figures 3 and 4.
- The Conclusion part is not sufficient to publish the present manuscript. However, this part should be rewritten.
- Please write in the same format the References: for example the references: 2, 3, 5, 21 are written in different modes. The references are adequate, necessary. I think the author must strengthen the References section with the references that use the same technique, to make the technique used more plausible, for instance:
Fabrication of Composite Filaments with High Dielectric Permittivity for Fused Deposition 3D Printing, Materials2017, 10(10), 1218. Stress singularity of symmetric free-edge joints with elasto-plastic behaviour, Comp. Mat. Sci., 52(1), p. 282-286, (2012).
If the authors take into account all these recommendations, then the manuscript deserves to be published.
Author Response
Point 1: The title is generic. Please modify the title.

Response 1: Thank you for your advice. The title has been modified to “Performance of 3D Printed Continuous Carbon Fiber Reinforced Plastics with Pressure”
Point 2: Please write lines 143-147.
Response 2: Thank you. The lines 143-147 have been modified to
“The flow chart for converting curve of pressure roller to line is shown in Figure 4a. The result of converting is shown in Figure 4b. The solid line is the curved path of pressure roller, the dotted line is the lined path of pressure path after conversion and the red area is the rolling area of the pressure roller after conversion. The curves are inside the red area, indicating that the rolling is sufficient. The steps for converting curves of pressure roller into lines are as follows:
Firstly, finding the starting point and the ending point of curve;
Secondly, calculating the max distance( ) of line with curve , ;
Thirdly, comparing the size of the with ( ); gradually accumulate, is the diameter of pressure roller;
Then, the is converted to while , and ;
Finally, follow the steps above to convert the curve to line starting from . “
Point 3: The Figure 3 and 4 are not clear. Author, please give a better quality to Figures 3 and 4.
Response 3: Thank you for your comments. The quality of Figures 3 and 4 have been adjusted.
Point 4: The Conclusion part is not sufficient to publish the present manuscript. However, this part should be rewritten.
Response 4: Thank you for your advices. The Conclusion part has been rewritten as follow:
“FDM has been investigated to a low-cost manufacturing method for fiber reinforced composites and there are many studies on this aspect currently. For the manufacture of CCFRPs parts, the pressure is essential. Without pressure, the CCFRPs between adjacent layers are adhered by the thermoplastic during the printing, CCFRPs molecular chains of two adjacent layers do not diffuse, and the degree of diffusion will affect the binding strength of the parts. The traditional and mature technology for manufacturing CCFRPs is AFP which uses consolidation roller and autoclave process to improve the quality of part. Compared to AFP, FDM is simple in design and operation but lack the ability to apply pressure and heat to the model. This work proposed a molding process for printing CCFRPs to combine the advantage of FDM and AFP. A novel 3D printing structure for CCFRPs is put forward in this article, which can heat and pressurize the CCFRPs during printing by pressure roller. Using this process, the CCFRPs are pressurized while printing to increase the bond between layers. Then the strength of printed model with this molding process is tested. The details as follows:
7 movements(X, Y, Z, F, C, R, L) in the molding process of printing CCFRP are coordinated to complete the printing. The pressure roller is rotated and lifted to apply pressure and heat to the part.
The path of pressure roller is different with nozzle. The curved path of pressure roller need to be converted to lined path to prevent jitter and improve efficiency. Then calculating the rotation angle of pressure roller based on the lined path. The accuracy of path processing of pressure roller has been proved by the experiment. The phenomenon shows that the pressure can enhance the bonding of CCF.
The tensile and bending strength of models printed by CCFRPs with different pressure have been tested to analyze the effect of pressure. The results show that the pressure has a significant effect on the reinforcement of CCF on PLA. The tensile strength and bending strength of model are enhanced to 644.8MPa, 401.24 with increasing the pressure. The results show that the error of each experiments is within 10%, which proves that the experiment is repeatable. However, the excessive pressure will destroy the path of CCF and surface quality of model and even lead to failure of printing.
It can be seen that the pressure is necessary for printing CCFRP with FDM. This work will provide the technical foundational for printing CCFRPs with FDM.”
Point 5: Please write in the same format the References: for example the references: 2, 3, 5, 21 are written in different modes. The references are adequate, necessary. I think the author must strengthen the References section with the references that use the same technique, to make the technique used more plausible, for instance:
Fabrication of Composite Filaments with High Dielectric Permittivity for Fused Deposition 3D Printing, Materials2017, 10(10), 1218. Stress singularity of symmetric free-edge joints with elasto-plastic behaviour, Comp. Mat. Sci., 52(1), p. 282-286, (2012).
Response 5: Thank you for your comments. The format of the References have been adjusted to the same. The references “Fabrication of Composite Filaments with High Dielectric Permittivity for Fused Deposition 3D Printing, Materials2017, 10(10), 1218” and “Stress singularity of symmetric free-edge joints with elasto-plastic behaviour, Comp. Mat. Sci., 52(1), p. 282-286, (2012)” have been used to strengthen the References section

Reviewer 2 Report
There are several issues that should be addressed in the manuscript before further consideration for publication.
1. Suggest to use standard ISO/ASTM terminology when describing the process
Goh et al. (2019), Process–structure–properties in polymer additive manufacturing via material extrusion: A review, Critical Reviews in Solid State and Materials Sciences, 1-21 Jahangir et al. (2019), Reinforcement of material extrusion 3D printed polycarbonate using continuous carbon fiber, Additive Manufacturing 28, 354-3642. What is the pressure exerted by the roller during the process?
3. How many specimens used for each experiment? Any repeatability/reliability test?
4. It appears that only a single layer parts are used for the tests? How is this 3D printing? Any proof of concepts with more complicated parts?
Author Response
Point 1: Suggest to use standard ISO/ASTM terminology when describing the process
Goh et al. (2019), Process–structure–properties in polymer additive manufacturing via material extrusion: A review, Critical Reviews in Solid State and Materials Sciences, 1-21 Jahangir et al. (2019), Reinforcement of material extrusion 3D printed polycarbonate using continuous carbon fiber, Additive Manufacturing 28, 354-364
Response 1: Thank you for your advice. I have described the process according to the standard ISO/ASTM52900:2015(E) terminology. The references “Goh et al. (2019), Process–structure–properties in polymer additive manufacturing via material extrusion: A review, Critical Reviews in Solid State and Materials Sciences, 1-21” and “Jahangir et al. (2019), Reinforcement of material extrusion 3D printed polycarbonate using continuous carbon fiber, Additive Manufacturing 28, 354-364” have been used to strengthen the References section.
Point 2: What is the pressure exerted by the roller during the process?
Response 2: Thank you. During the printing process, the movement accuracy is 0.1mm and the jitter is large. The precise value of the pressure is not easy to measure. In this manuscript, increasing the distance(d) of pressure roller below the nozzle to indicate the increase in pressure, as show below. This article studies the effect of pressure on printing CCFRPs and the effect of increased pressure on performance.
Point 3: How many specimens used for each experiment? Any repeatability/reliability test?
Response 3: Thank you. Three samples were tested in each experiment in this article. The results show that the error of each experiment is within 10%, which proves that the experiment is repeatable.
Point 4: It appears that only a single layer parts are used for the tests? How is this 3D printing? Any proof of concepts with more complicated parts?
Response 4: Thank you. The path experiments of pressure roller adopt a single layer, which can clearly observe the experimental phenomenon. The samples of tensile and bending experiments are not a single layer and the sample thickness is 2mm, about 10 layers, as shown below.

Reviewer 3 Report
In this paper, the authors explore the use of post-printing pressure on carbon-fiber reinforced FDM films. This is an interesting topic, but the paper is incomplete and the authors do not make a good case for the novelty of the method nor its mechanics. The improvement in performance is quite dramatic and I would like to see more detail about why and how these results are obtained. I suggest that the manuscript should be returned to the authors for a major revision to address the comments and points given below.
1. The paper is very short for a materials processing method paper and reads more like a communication or technical note. The authors should expand it and provide more detail about the setup, experiments, etc. The presentation of the tests should focus on experimental repeatibility and how the work fits within existing methods. More focus should also be given in the introduction on exactly what the new and original contribution of this paper is and why it is valid to present this work as a research paper even though there is very little to no theoretical development (and hence why this seems like a technical note and not a research paper to me).
2. The literature review focuses on collecting material property data from previous works, which is excellent, but the printing methods also need more space here.
3. The focus of the paper seems to be too much on specific material properties and not on the processing method presented. This work is not extensive or formal enough to be considered a valid material characterization, so the major contribution of the work is the rolling method. This is very good and a useful contribution (once fully developed), but this does not seem to be given enough prominence in this paper. The authors should spend more time at the beginning of the paper discussing the need for the pressure, why it can help, etc.
4. The presented method of rolling seems (from looking at your data) seems to help by removing voids between the material elements and assisting with the more complete polymerization of the material on cooling. This (or whatever the authors decide is most appropriate) needs discussion and presentation. If this in indeed the case that this is the basic mechanism, this method presented here is likely valid for other materials and composites, expanding the value and usefulness of the method. If the authors have no background in polymer chemistry, it would be helpful to cite some papers in this area to explain what is going on. Otherwise the authors should present this.
5. The authors need to explain in more detail the impact of poor printing (as seen in Figure 9) and why they accepted samples like this for testing. If I was conducting this experiment, I would have rejected these samples and updated my settings until I got films without obvious voids. These samples basically are so loosely connected that they are almost a bundle of fibers and not a film. I understand that this is often how these materials behave under normal conditions, but it certainly needs explanation and clarification. The samples that were actually tested are better.
7. Was the pressure on the samples based on (1) pressure value or (2) distance from the printbed? This matters a lot for understanding the results.
7. The authors focus on films for this study and not on bulk materials. I think this is ok for a first study on the topic, but I think the authors will find less dramatic results for bulk materials (especially when the bulk is large enough for a plane strain condition in the material) due to sliding planes, fiber pullout, etc. This does not need to be formally addressed here, but the authors do need to discuss it and perhaps modify the paper title to more clearly represent that the study is about very thin parts under essentially pure plane stress conditions.
8. Many of the figures are low quality and need to be replaced with higher-resolution images. In addition, Figure 6 seems to be quite distorted and some details are hard to understand. There is no page limit for this journal, so please use large and clear images in order to best represent the study.
9. There are two confounding factors in this experiment that are not explored and need to be addressed: (1) the amount of time between printing and the roller pressure (this is material dependent and PLA is very sensitive to it - rapidly cooled PLA will be more amorphous and slowly cooled PLA will be more crystalline.) and (2) the conditions of the printbed - was a heated bed used (for PLA films, a heated bed almost always means a crystalline polymer structure, which may have some kind of influence on your results)? Was any kind of epoxy or surfacing used besides the blue painter's tape (if so, how was is removed before testing)? Is the tape used paper-based or polymer-based? If paper, no issues - but if polymer, was any tape residue left on the samples during testing?
10. The authors should at least suggest a method for doing a more complete and rigorous study using this method - preferably with a designed experiment (factorial analysis, Taguchi, etc.) for 5-10 replications (depending on whether ASTM or ISO standards are used). As is, it is ok to prove the concept but should certainly be given more formal attention in the future. I am especially interested in seeing the variance between replications of these tests and an analysis of how much of this is due to the improved polymer matrix after rolling and how much is from the fibers.
Please consider these comments. I look forward to seeing a revised version of this paper.
Author Response
Point 1: The paper is very short for a materials processing method paper and reads more like a communication or technical note. The authors should expand it and provide more detail about the setup, experiments, etc. The presentation of the tests should focus on experimental repeatibility and how the work fits within existing methods. More focus should also be given in the introduction on exactly what the new and original contribution of this paper is and why it is valid to present this work as a research paper even though there is very little to no theoretical development (and hence why this seems like a technical note and not a research paper to me).
Response 1: Thank you for your comments. The article adds experimental parameters settings as shown below:
“3.2. Printing process parameters
Printing the CCFRPs is different from printing the PLA. The feeding speed must be consistent with the nozzle moving speed to avoid the accumulation and breaking of CCFRPs and the speed must be slow to ensure the rolling effect of pressure roller when printing the CCFRPs. Setting the printing parameters is crucial for printing CCFRPs and CURA is used to slice the part in this article. The multi-process parameters of printing are specified as the Table 1.”
Table 1 The parameters of printing CCFRPs
Parameter Amount
Nozzle diameter 0.8mm
Material diameter 0.58mm
Feeding speed 20mm/min
Moving speed 20mm/min
Rotating speed 2rad/s
Layer thickness 0.2mm
Infill percentage 100%
Fiber content 10.3%
Extrusion temperature 200°C
Pressure roller temperature 180°C
Three samples were tested in each experiment in this article. The results show that the error of each experiment is within 10%, which proves that the experiment is repeatable.
The new and original contribution of this article has been modified in the introduction as shown below:
“To solve the above problem, a novel 3D printing structure for CCFRPs is put forward in this article, which can heat and pressurize the CCFRPs during printing by pressure roller. Using this process, the CCFRPs are pressurized while printing to increase the bond between layers. The path processing of pressure roller is designed and verified to ensure the feasibility and rationality of algorithm. Then, the testing of printing CCFRPs is performed and the mechanical properties of parts under pressure are analyzed. In this paper, a pressure function is added to 3D printing for CCFRPs, and a matching pressure roller path processing algorithm is studied. The results show that the pressure has an obvious improvement on the strength of parts printed by CCFRPs. This article will contribute a new reference and data to the application of 3D printing for CCFRPs.”
Point 2: The literature review focuses on collecting material property data from previous works, which is excellent, but the printing methods also need more space here.
Response 2: Thank you for your advices. In the introduction, more detailed descriptions of the two methods of printing CCFRPs are as follows “As shown in Figure 1a, the CCFs were impregnated with thermoplastic in the nozzle and the CCFRPs can be printed in a single step [21]. The CCFs and thermoplastic are compounded inside the nozzle, the CCFs are brought out while the thermoplastic is extruded depending on the viscosity of the thermoplastic.” And “The other strategy is to manufacture the CCFRPs and then print parts with the CCFRPs on a FDM printer, which is a multi-step process, as shown in Figure 1b. The CCFs and thermoplastic are compounded into CCFRPs on a specific material preparation machine which can control the diameter and fiber content of CCFRPs. Then the prepared CCFRPs are used to print the parts on the 3D printer, and the process parameters of the 3D printer need to match the continuity characteristics of the CCFRPs.”
Point 3: The focus of the paper seems to be too much on specific material properties and not on the processing method presented. This work is not extensive or formal enough to be considered a valid material characterization, so the major contribution of the work is the rolling method. This is very good and a useful contribution (once fully developed), but this does not seem to be given enough prominence in this paper. The authors should spend more time at the beginning of the paper discussing the need for the pressure, why it can help, etc.
Response 3: Thank you for your advices. In the introduction, a more detailed description of the necessary for pressure for printing CCFRPs is as follows:
“Without pressure, the CCFRPs between adjacent layers are adhered by the thermoplastic during the printing, CCFRPs molecular chains of two adjacent layers do not diffuse, and the degree of diffusion will affect the binding strength of the parts. For the manufacture of CCFRPs parts, the pressure is essential. Lee et al. presented a model in the manufacturing process of semicrystal line thermoplastic matrix composites and studied the effect of pressure on the bonding of two adjacent layers of composites[34].”
Point 4: The presented method of rolling seems (from looking at your data) seems to help by removing voids between the material elements and assisting with the more complete polymerization of the material on cooling. This (or whatever the authors decide is most appropriate) needs discussion and presentation. If this in indeed the case that this is the basic mechanism, this method presented here is likely valid for other materials and composites, expanding the value and usefulness of the method. If the authors have no background in polymer chemistry, it would be helpful to cite some papers in this area to explain what is going on. Otherwise the authors should present this.
Response 4: Thank you for your advices. Related papers are cited in this paper to prove that the interlayer bonding strength is related to the diffusion of molecular chains and the better the interface bonding strength with increasing pressure on the CCFRPs.
“Wool et al. presented a theory of crack healing in polymers and the surface rearrangement stage effects the diffusion initiation function and topological features of the interface[35]. Kim et al. presented a microscopic theory of the diffusion and randomization stages based on the reptation model of chain dynamics by de Gennes and proved that the interlayer bonding strength is proportional to the molecular chain diffusion [36]. Song et al. presented the interlaminar intimate contact model and the molecular chain diffusion mode and the experimental results found the degree of the interlaminar intimate could reach 1 when the pressure of roller reaches 1500 N[37]. The effects of pressure on CCFRPs in the above studies are all based on AFP of manufacturing CCFRPs, but less research on the preparation of CCFRPs parts on 3D printing.”
Point 5: The authors need to explain in more detail the impact of poor printing (as seen in Figure 9) and why they accepted samples like this for testing. If I was conducting this experiment, I would have rejected these samples and updated my settings until I got films without obvious voids. These samples basically are so loosely connected that they are almost a bundle of fibers and not a film. I understand that this is often how these materials behave under normal conditions, but it certainly needs explanation and clarification. The samples that were actually tested are better.
Response 5: Thank you for your comments. The explanation of the poor printing in the article is as follows:
“The schematic diagram of the pressurized printing CCFRPs is shown in Figure 9c. The distance between the nozzle and the platform is 0.2mm to keep that the CCFRPs are kept continuous and adhered to the platform during printing. The CCFRPs are laid on the platform by their own gravity and the adhesion between the thermoplastic and the platform. The part printed without pressure is shown in Figure 9a. The CCFRPs in one layer are bonded together by PLA. There are gaps between the CCFRPs without applying pressure on part, resulting in reducing adhesion of CCF. The pressure roller is lower than the nozzle and can press the parts. Pressure allows the CCF to collapse and expanded, extruding the excess PLA from CCFRPs and applying the excess PLA on the expanded CCF. Figure 9b shows the part printed in accordance with the path shown in the Figure 5. This process can increase the cross-linking of CCF and the strength of model, and the validity of algorithm of pressure roller is verified. Pressurization is a compaction process which can reinforce the cross-linking of inter-laminar CCFRPs and the adhesion of adjacent layer of CCFRPs, so the strength of model can be increased. ”
Figure 9. The printing model without pressure(a) and with pressure(b), (c) the schematic diagram of the pressurized printing CCFRPs.
Point 6: Was the pressure on the samples based on (1) pressure value or (2) distance from the printed? This matters a lot for understanding the results.
Response 6: Thank you. The precise value of the pressure is not easy to measure. In this manuscript, increasing the distance(d) of pressure roller below the nozzle to indicate the increase in pressure, as show below.
Point 7: The authors focus on films for this study and not on bulk materials. I think this is ok for a first study on the topic, but I think the authors will find less dramatic results for bulk materials (especially when the bulk is large enough for a plane strain condition in the material) due to sliding planes, fiber pullout, etc. This does not need to be formally addressed here, but the authors do need to discuss it and perhaps modify the paper title to more clearly represent that the study is about very thin parts under essentially pure plane stress conditions.
Response 7: Thank you for your comments. The title has been modified to “Performance of 3D Printed Continuous Carbon Fiber Reinforced Plastics with Pressure” The path experiments of pressure roller adopt a single layer, which can clearly observe the experimental results. The samples of tensile and bending experiments are not a single layer. Sample thickness is 2mm, about 10 layers, as shown below.
Point 8: Many of the figures are low quality and need to be replaced with higher-resolution images. In addition, Figure 6 seems to be quite distorted and some details are hard to understand. There is no page limit for this journal, so please use large and clear images in order to best represent the study.
Response 8: Thank you for your advices. I have revised them, large and clear images have been used to best represent the study.
Point 9:There are two confounding factors in this experiment that are not explored and need to be addressed: (1) the amount of time between printing and the roller pressure (this is material dependent and PLA is very sensitive to it - rapidly cooled PLA will be more amorphous and slowly cooled PLA will be more crystalline.) and (2) the conditions of the printbed - was a heated bed used (for PLA films, a heated bed almost always means a crystalline polymer structure, which may have some kind of influence on your results)? Was any kind of epoxy or surfacing used besides the blue painter's tape (if so, how was is removed before testing)? Is the tape used paper-based or polymer-based? If paper, no issues - but if polymer, was any tape residue left on the samples during testing?
Response 9: Thank you for your comments. The section “3.2 Printing process parameters” is added in this paper, and the printing speed is 20mm/min(slowly speed) to ensure the effect of pressure. This article uses air cooling(slowly cooled) to make PLA more crystalline. The printbed in this article is not heated. No epoxy or surfacing for the parts. The tape use paper-based and no tape residue left on the samples during testing.
Point 10: The authors should at least suggest a method for doing a more complete and rigorous study using this method - preferably with a designed experiment (factorial analysis, Taguchi, etc.) for 5-10 replications (depending on whether ASTM or ISO standards are used). As is, it is ok to prove the concept but should certainly be given more formal attention in the future. I am especially interested in seeing the variance between replications of these tests and an analysis of how much of this is due to the improved polymer matrix after rolling and how much is from the fibers.
Response 10: Thank you for your comments. Three samples were tested in each experiment in this article. The results show that the error of each experiment is within 10%, which proves that the experiment is repeatable.

Round 2
Reviewer 2 Report
There are still some issues in the manuscript that should be addressed before further consideration for publication.
The nozzle distance to the platform is kept at 0.2 mm consistently? So how many layers are needed to produce, for example, the tensile samples which is 2 mm? Is it 10 layers? The test parameters and manufacturing parameters need to be more clearly defined.
From the test samples images, it appears that there are voids within the sample. Are the process parameter optimised? Any characterisation done on this? For example, the usage of CT scans?
Author Response
Point 1: The nozzle distance to the platform is kept at 0.2 mm consistently? So how many layers are needed to produce, for example, the tensile samples which is 2 mm? Is it 10 layers? The test parameters and manufacturing parameters need to be more clearly defined.
Response 1: Thank you for your comments. The distance between the nozzle and the upper surface of part is kept at 0.2mm, as shown below. The test sample is 10 layers, and the thickness of sample without pressure is 2mm, the thickness of the sample under pressure is less than 2mm.
The test parameters and manufacturing parameters are revised as below: “Printing the CCFRPs is different from printing the PLA. The feeding speed must be consistent with the nozzle moving speed to avoid the accumulation and breaking of CCFRPs and the speed must be slow to ensure the rolling effect of pressure roller when printing the CCFRPs, the feeding speed and the moving speed is 20mm/min. The material diameter is 0.58mm, layer thickness is set to 0.2mm to increase adhesion between layers. In this paper, 1k carbon fiber is used to prepare the composite material, and the fiber content is 10.3%. Setting the printing parameters is crucial for printing CCFRPs and CURA is used to slice the part in this article. The multi-process parameters of printing are specified as the Table 1.”
Table 1 The parameters of printing CCFRPs
|
Parameter |
Amount |
|
Nozzle diameter |
0.8mm |
|
Material diameter |
0.58mm |
|
Feeding speed |
20mm/min |
|
Moving speed |
20mm/min |
|
Rotating speed |
2rad/s |
|
Layer thickness |
0.2mm |
|
Infill percentage |
100% |
|
Fiber content |
10.3% |
|
Extrusion temperature |
200°C |
|
Pressure roller temperature |
180°C |
Point 2: From the test samples images, it appears that there are voids within the sample. Are the process parameter optimised? Any characterisation done on this? For example, the usage of CT scans?
Response 2: Thank you. There are voids in the middle of the sample due to the pressure. Under the action of the pressure roller, the CCFRPs will be shifted to the outside, so that a gap appears in the middle of the model, as shown in the figure below. The left picture is a schematic diagram without pressure, and the right picture is a schematic diagram with pressure. The existence of voids in the sample is inevitable, and the parameters have been optimized to minimize the voids in this article. The project team does not have the conditions for CT scans, thank you.

Reviewer 3 Report
The authors have addressed all my comments in a thorough and professional manner. I am satisfied with the current version and I recommend it be considered for acceptance.
Author Response
Thank you very much for your great efforts on our manuscript.
Round 3
Reviewer 2 Report
NA